# Sugar sensation and mechanosensation in the egg-laying preference shift of *Drosophila suzukii*

Wanyue Wang, Hany KM Dweck, Gaëlle JS Talross, Ali Zaidi, Joshua M Gendron, John R Carlson*

Department of Molecular, Cellular and Developmental Biology, Yale University, New Haven, United States

**Abstract** The agricultural pest *Drosophila suzukii* differs from most other *Drosophila* species in that it lays eggs in ripe, rather than overripe, fruit. Previously, we showed that changes in bitter taste sensation accompanied this adaptation (Dweck et al., 2021). Here, we show that *D. suzukii* has also undergone a variety of changes in sweet taste sensation. *D. suzukii* has a weaker preference than *Drosophila melanogaster* for laying eggs on substrates containing all three primary fruit sugars: sucrose, fructose, and glucose. Major subsets of *D. suzukii* taste sensilla have lost electrophysiological responses to sugars. Expression of several key sugar receptor genes is reduced in the taste organs of *D. suzukii*. By contrast, certain mechanosensory channel genes, including *no mechanoreceptor potential C*, are expressed at higher levels in the taste organs of *D. suzukii*, which has a higher preference for stiff substrates. Finally, we find that *D. suzukii* responds differently from *D. melanogaster* to combinations of sweet and mechanosensory cues. Thus, the two species differ in sweet sensation, mechanosensation, and their integration, which are all likely to contribute to the differences in their egg-laying preferences in nature.

*For correspondence: john.carlson@yale.edu

**Competing interest:** The authors declare that no competing interests exist.

## Editor's evaluation

The agricultural pest *Drosophila suzukii* displays an interesting and costly evolutionary adaptation of preferring ripe to overripe foods, as compared with the well-studied fruit fly *Drosophila melanogaster*. This study identifies mechanisms that may contribute to this evolutionary shift, including changes in the expression levels of gustatory sweet receptors and mechanoreceptors, and altered electrophysiological responses to sugars. Additional studies involving genetic perturbations in *D. suzukii* are needed in the future to determine the extent to which observed changes contribute to the evolution of substrate preference for egg laying.

## Introduction

*Drosophila suzukii*, commonly known as the spotted wing *Drosophila*, is a major agricultural pest of soft fruits, including strawberries, raspberries, and blueberries (*Burrack et al., 2013*; *Lee et al., 2011*; *Mazzi et al., 2017*). It invaded the continental United States in 2008 and is now found in at least 52 countries worldwide (*Andreazza et al., 2017*; *Calabria et al., 2012*; *Dos Santos et al., 2017*; *Hauser, 2011*; *Ørsted et al., 2019*). Efforts to control its damage to fruit production have relied largely on insecticides, and improved means of control are critically needed.

 *D. suzukii* is destructive due to its unusual egg-laying preference. Most *Drosophila* species, including *Drosophila melanogaster*, prefer to lay eggs on fermenting fruits. By contrast, *D. suzukii* has an egg-laying preference for ripe, intact fruits (*Cini et al., 2012*; *Figure 1—figure supplement 1*). *D. suzukii*

females have an enlarged saw-like ovipositor that can pierce the skin of intact fruits and insert eggs underneath (*Poyet et al., 2015*). However, little is known about the sensory mechanisms underlying their different egg-laying preference. *D. suzukii* provides an excellent opportunity for comparative studies of how sensory systems evolve, taking advantage of the vast accumulated knowledge and genetic tools of *D. melanogaster*.

*D. melanogaster* females select egg-laying sites by evaluating many cues, which inform them of nutrients, microbes, predators, and other flies (*Ebrahim et al., 2015*; *Lin et al., 2015*; *Stensmyr et al., 2012*). Multiple sensory modalities are used: long-range localization of appropriate sites relies mainly on olfaction and vision, whereas close-range decisions rely on contact-dependent gustatory and mechanosensory signals (*Markow, 2019*).

As fruits progress through stages of ripening and fermentation, they undergo many changes, including alterations in softness, sugar content, acidity, and odor (*Dudley, 2004*; *Hidalgo et al., 2013*; *Paul and Pandey, 2014*). A priori, any of these changes could serve as fruit stage indicators for the fly, and alterations in the sensation of any could contribute to the unusual egg-laying preference of *D. suzukii*. A pioneering study showed elegantly that changes in olfactory and mechanosensory responses contribute to the shift, but left open the possibility that other changes might contribute as well (*Karageorgi et al., 2017*).

Taste systems evaluate the nutrient content and toxicity of potential food sources, and gustation is crucial in the egg-laying decisions of a variety of insect species (*Joseph and Carlson, 2015*; *Montell, 2021*; *Scott, 2018*). Some gustatory cues are thought to potentially influence *D. suzukii*'s behavior. *D. suzukii*'s egg-laying preference has been found to correlate with the phosphorus content of fruits (*Olazcuaga et al., 2019*), and the protein:carbohydrate ratio may be another cue (*Silva-Soares et al., 2017*; *Young et al., 2018*). A recent study found a difference between *D. suzukii* and *D. melanogaster* in egg-laying preference for high sucrose concentrations (*Durkin et al., 2021*). An extensive behavioral, electrophysiological, and molecular analysis of taste organs recently showed that *D. suzukii* and *D. melanogaster* sense bitter compounds differently (*Dweck et al., 2021*), inviting a comparable comparison of the sensation of other salient taste cues.

Here, we investigate the sensation of sugars in *D. suzukii* and its contribution to the shift in egg-laying preference toward ripe fruit. Sugars are ubiquitous in fruits, are a major energy source for flies, and undergo changes in concentration during fruit ripening. As a fruit becomes increasingly over-ripe and its surface deteriorates, the accessibility of its sugars to a fly may also change. We provide evidence that a change in sugar sensation contributes to the difference in oviposition preference between *D. suzukii* and *D. melanogaster*. We show that *D. suzukii* has a weaker egg-laying preference

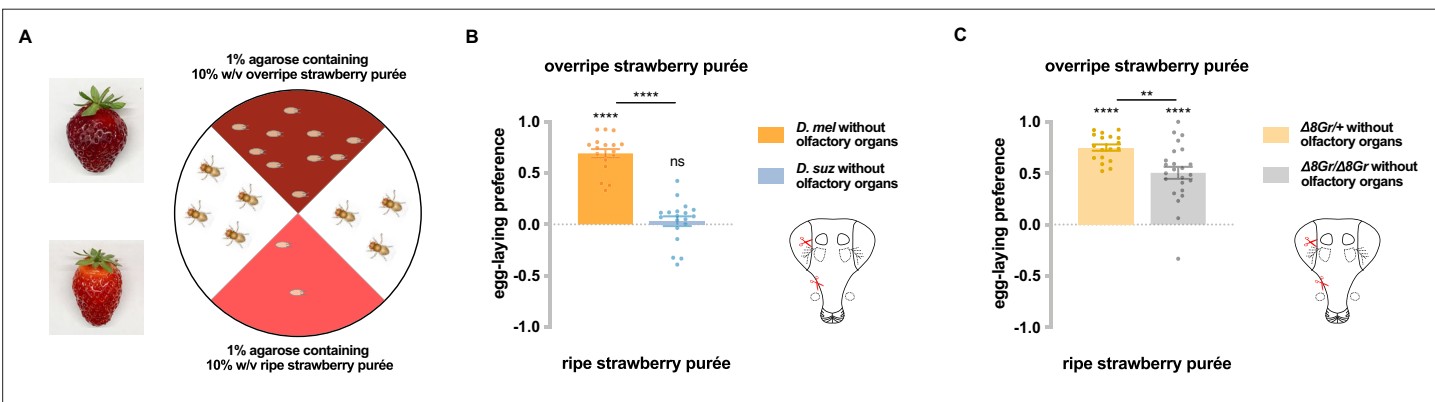

**Figure 1.** Taste contributes to the difference in egg-laying preference between *D. suzukii* and *D. melanogaster*. (**A**) Two-choice egg-laying paradigm. Female flies (n=4–10 flies per plate) whose olfactory organs had been removed were allowed to lay eggs in the dark. The preference index was calculated as (#eggs in overripe purée - #eggs in ripe purée)/total #eggs. (**B**) Egg-laying preferences of the two species, without olfactory organs. Each egg-laying preference index was compared to 0 using the Wilcoxon signed-rank test. The Mann-Whitney test was used to compare the preference indices between species. n=18–20. ns, not significant; *p<0.05; ****p<0.0001. (**C**) Egg-laying preference of females homozygous mutant for eight sugar receptor genes and of control females heterozygous for the eight mutations. n=18–20. Error bars are SEM.

The online version of this article includes the following figure supplement(s) for figure 1:

**Figure supplement 1.** *D. suzukii* has an egg-laying preference for ripe strawberries and for ripe strawberry purée, while *D. melanogaster* prefers overripe strawberries and overripe strawberry purée.

than *D. melanogaster* for sweeter substrates, that a number of *D. suzukii* taste sensilla have lost electrophysiological responses to sugars, and that a number of sugar receptors are expressed at lower levels in the taste organs of *D. suzukii* than in *D. melanogaster*. We confirm earlier reports that *D. suzukii* and *D. melanogaster* have different preferences for substrate stiffness (***Durkin et al., 2021***; ***Guo et al., 2020***; ***Karageorgi et al., 2017***) and find that *D. suzukii* has higher expression of the mechanosensory channel *no mechanoreceptor potential C* (*nompC*) in its taste organs. We investigate the integration of sugar and mechanosensory cues and find that *D. suzukii* and *D. melanogaster* respond differently to combinations of sweetness and stiffness in egg-laying behavior. Thus, *D. suzukii* and *D. melanogaster* differ in sweet sensation, mechanosensation, and their integration, all of which are likely to contribute to their natural preferences for ripe and overripe fruits, respectively.

## Results

### Taste contributes to the difference in egg-laying preference between the two species

We first wanted to confirm that differences in the taste responses of *D. melanogaster* and *D. suzukii* contribute to their differences in egg-laying preference for ripe vs. overripe strawberry. We tested their preferences in a two-choice egg-laying paradigm in which the flies could lay eggs on purées of either ripe or overripe strawberry (***Figure 1A***). To minimize visual cues, the assay was performed in the dark; to minimize mechanosensory cues, equivalent concentrations of agarose were added to each purée; to minimize olfactory cues, we surgically removed the olfactory organs—the antennae and maxillary palps—from the flies. Deprived of these other cues, *D. melanogaster* showed a robust preference for the overripe purée, while *D. suzukii* did not (***Figure 1B***). The simplest interpretation of this result is that taste contributes to the difference in egg-laying preference between the two species.

Does the strong preference of *D. melanogaster* for overripe strawberry depend on sugar sensation? We took advantage of an octuple mutant in which eight of nine *Gr* sugar receptor genes are mutated (***Ahn et al., 2017***; ***Yavuz et al., 2014***). After their olfactory organs had been removed, these mutant flies showed a lower preference for overripe strawberry purée than control flies whose olfactory organs had also been removed (***Figure 1C***). These results suggest that sugar sensation contributes to the preference of *D. melanogaster* for overripe fruit purée.

### *D. suzukii* shows a weaker egg-laying preference than *D. melanogaster* for sweeter substrates

To investigate whether *D. suzukii* and *D. melanogaster* differ in their response to sugars, we first used a single-fly two-choice egg-laying preference paradigm. Flies can choose to lay eggs on either of two agarose substrates containing different sugar concentrations: one with 100 mM sugar, and the other with either 0 mM, 10 mM, 30 mM, or 60 mM concentrations of the same sugar (***Figure 2A***). A preference index was calculated based on the number of eggs on each substrate. Sucrose, fructose, and glucose, the main sugars in most fruits, were each tested.

Both *D. suzukii* and *D. melanogaster* preferred the medium with 100 mM sucrose to that with no sucrose, but the preference of *D. suzukii* was weaker than that of *D. melanogaster* (***Figure 2B***), consistent with a recent study that used higher concentrations (***Durkin et al., 2021***). When choosing between 100 mM and 10 mM sucrose, *D. melanogaster* again showed a strong preference for the higher concentration, but *D. suzukii* showed little if any preference. And when choosing between 100 mM and 30 mM or between 100 mM and 60 mM sucrose, *D. suzukii* again exhibited a weaker preference than *D. melanogaster*.

Fructose also elicited weaker preferences from *D. suzukii* than *D. melanogaster*, in each of the four preference tests (***Figure 2C***). Both species preferred 100 mM fructose to plain agarose, but the preference of *D. suzukii* was weaker. Whereas *D. melanogaster* preferred the higher concentration of fructose in the other three tests, *D. suzukii* showed no preference.

Glucose showed similar results: in every comparison, *D. suzukii* showed a weaker preference for the higher sugar concentration than *D. melanogaster* (***Figure 2D***). Again, when the concentration differences were less extreme, *D. suzukii* showed no preference.

To test the possibility that the preferences of flies for high-sugar concentrations were exclusively due to a preference for high osmolarity, we set up a choice between 100 mM sucrose and 100 mM

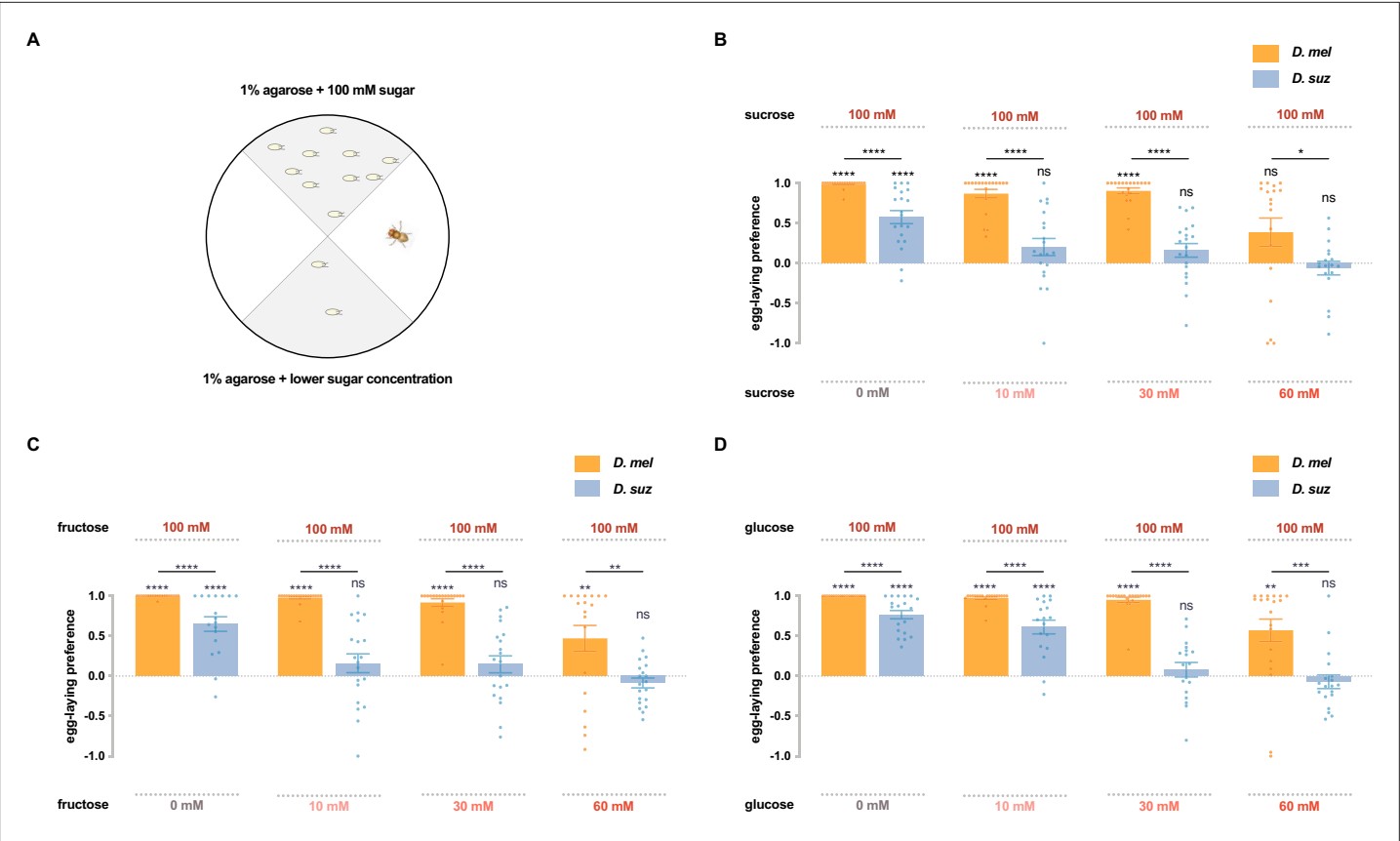

**Figure 2.** *D. suzukii* shows a weaker egg-laying preference than *D. melanogaster* for sweeter substrates. (**A**) The single-fly egg-laying preference paradigm. The preference index is calculated as (#eggs in higher sugar concentration - #eggs in lower sugar concentration)/total #eggs. (**B–D**) Preference indices for the indicated concentrations of (**B**) sucrose, (**C**) fructose, and (**D**) glucose. Each egg-laying preference index was compared to 0 using the Wilcoxon signed-rank test. The Mann-Whitney test was used to compare the preference indices between species. n=18–20. ns, not significant; *p<0.05; **p<0.01; ***p<0.001; ****p<0.0001. Error bars are SEM.

The online version of this article includes the following figure supplement(s) for figure 2:

**Figure supplement 1.** Preferences for high-sugar concentrations are not exclusively due to a preference for high osmolarity.

sorbitol, a sugar alcohol that is generally considered tasteless to flies (*Dahanukar et al., 2007*). If flies were responding uniquely to osmolarity, then one would predict that flies would show no preference between 100 mM sucrose and 100 mM sorbitol. In fact, flies of both species showed strong preferences to 100 mM sucrose (*Figure 2—figure supplement 1*). Moreover, the preferences were the same as those between 100 mM sucrose and plain medium, as if they are insensitive to the osmolarity of sorbitol. The simplest interpretation of these results is that *D. suzukii* and *D. melanogaster* differ in their response to sweetness.

In conclusion, *D. suzukii* showed a weaker egg-laying preference for sweeter substrates than *D. melanogaster*.

## Major subsets of *D. suzukii* taste sensilla have lost sugar responses

As species evolve and adapt to new environments, changes can occur either in sensory neurons or in the circuits that they drive. We wondered if the shifts we have found in the taste behavior of *D. suzukii* could be explained at least in part by changes in peripheral physiology.

There are 31 taste sensilla in the labellum, the primary taste organ of the *D. melanogaster* head: 11 small (S) sensilla, 9 large (L) sensilla, and 11 intermediate (I) sensilla (*Weiss et al., 2011*). The sensillum repertoire of *D. suzukii* is similar in its spatial organization but has lost two S sensilla and two I sensilla (*Figure 3A*; *Dweck et al., 2021*). We examined the electrophysiological responses (*Figure 3B–K*) of

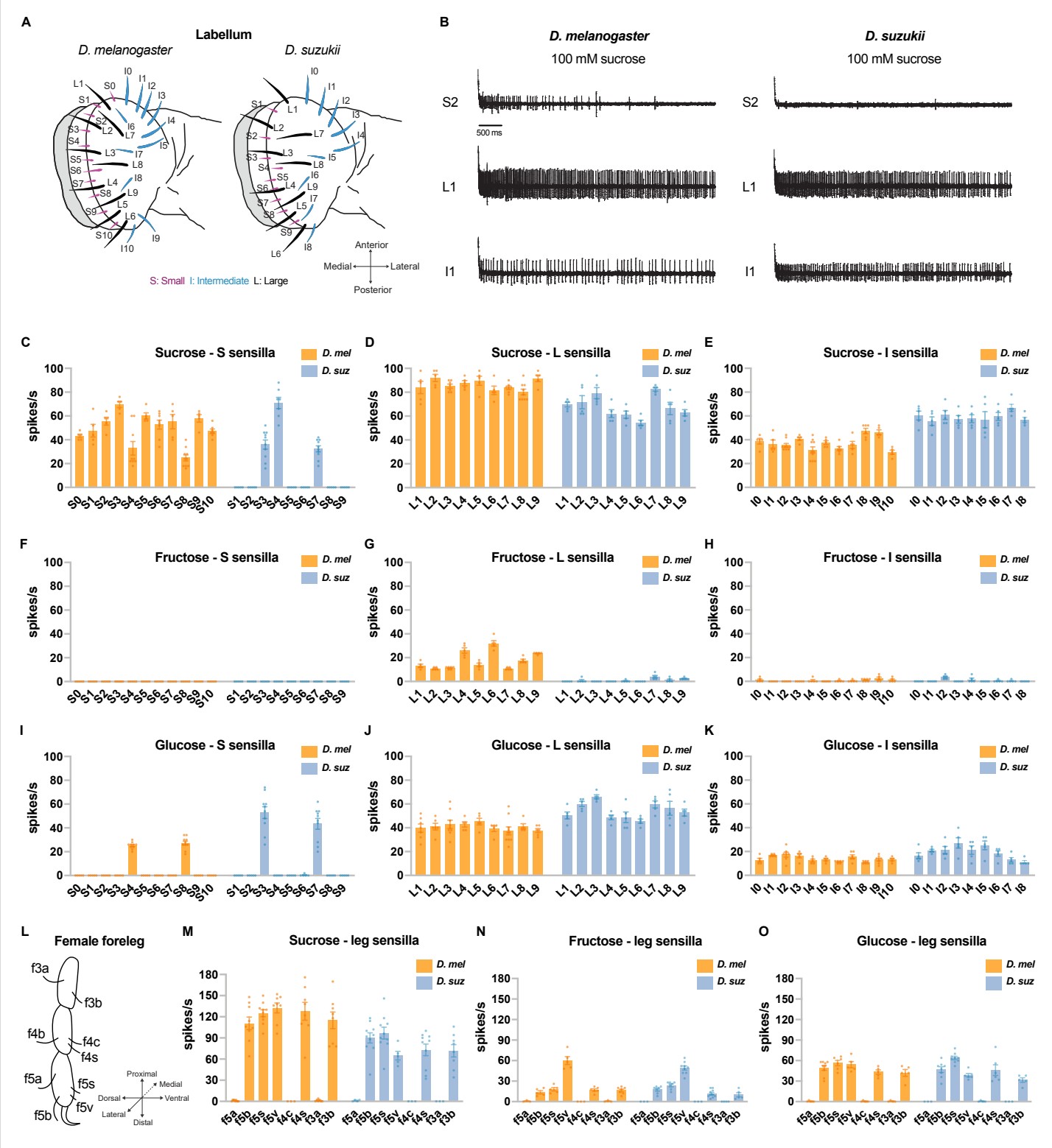

**Figure 3.** Electrophysiological responses to sugars. All sugars were tested at 100 mM concentrations. (**A**) Taste sensilla of the labellum. Figure 3A is reproduced from Figure 3D by *Dweck et al., 2021*. (**B**) Sample traces from the indicated sensilla. (**C–K**) Responses to the indicated sugar of each sensillum of each morphological class: S=small; L=large; I=intermediate. (**L**) Taste sensilla on the three distal tarsal segments of the female foreleg; the map is the same for *D. suzukii* and *D. melanogaster*. (**M–O**) Responses to the indicated sugar of each of the indicated tarsal sensilla. n=5-10 for responses≥5 spikes/s; otherwise, n=3-10. Error bars are SEM.

*Figure 3 continued on next page*

*Figure 3 continued*

The online version of this article includes the following figure supplement(s) for figure 3:

**Figure supplement 1.** Dose dependence of sucrose responses.

all taste sensilla on the labellum of both species to 100 mM concentrations of sucrose, fructose, and glucose.

A number of *D. suzukii* labellar sensilla have lost sugar responses. Sucrose elicited responses from all 11 of the S sensilla in *D. melanogaster*, but only 3 of the S sensilla in *D. suzukii* (*Figure 3B* top traces, *Figure 3C*). Fructose evoked responses of more than 10 spikes/s from all 9 of the L sensilla in *D. melanogaster,* but not from any of their *D. suzukii* counterparts (*Figure 3G*). We note that in *D. melanogaster*, the electrophysiological responses to fructose are weaker than those to sucrose. Glucose responses were comparable in the two species (*Figure 3I–K*).

We also carried out recordings from taste sensilla on the forelegs, which mediate oviposition preferences for at least some types of substrates in *D. melanogaster* (*Chen and Amrein, 2017*). We found that the number and spatial organization of taste sensilla on the three distal tarsal segments of the *D. suzukii* female foreleg are stereotyped and similar to those in *D. melanogaster* (*Figure 3L*). When tested with sucrose, fructose, and glucose at 100 mM concentrations, five *D. suzukii* female foreleg sensilla responded; other sensilla did not (*Figure 3M–O*). This pattern of responses was the same as that observed in *D. melanogaster* (*Ling et al., 2014*).

Among the sensilla that responded strongly to 100 mM sugar concentrations in both species, there could be differences between species in their dose-response relationships. We tested the labellum sensillum L8 and the leg sensillum f5s of both species at a series of sucrose concentrations and found that the dose-response relationships were comparable, although not identical; for example, the responses of f5s were lower in *D. suzukii* at the higher concentrations (*Figure 3—figure supplement 1A, B, D and E*). As a byproduct, this analysis offered an opportunity to examine the relationship between physiology and behavior. These sensilla, in both species, give distinguishable physiological responses to 1 mM sucrose vs. 10 mM sucrose and to 10 mM sucrose vs. 60 mM sucrose (*Figure 3—figure supplement 1C and F*). Likewise, both species could also distinguish between these concentrations behaviorally (*Figure 3—figure supplement 1G and H*). By contrast, flies of neither species distinguished behaviorally between 60 mM and 100 mM sucrose, and physiologically, the sensilla we examined did not distinguish between these concentrations in three of four cases (L8 of *D. suzukii* and f5s of both species; *Figure 3—figure supplement 1C and F*).

In summary, we found differences in the physiological responses of the two species to sugars. A major subset of S sensilla have lost response to sucrose in *D. suzukii*, and L sensilla have lost response to fructose. These losses could contribute to the weaker egg-laying preference of *D. suzukii* for sweeter substrates.

## Reduced expression of sugar receptor genes in the leg and labellum of *D. suzukii*

We wondered if the taste organs of *D. suzukii* and *D. melanogaster* differed in their expression of sugar receptor genes. We first constructed leg transcriptomes for female forelegs of both species. Rather than use entire legs, we dissected them so as to collect the tibia and tarsal segments, which contain taste sensilla, and to exclude other segments, which contain a large mass of muscle tissue. Four biological replicates were analyzed from each species, with each replicate containing the tibia and tarsi of 600 legs.

By focusing our analysis on leg segments containing taste sensilla, we were able to detect the expression of 13 *Gustatory receptor (Gr)* genes in the leg of *D. melanogaster* (*Figure 4—figure supplement 1*, *Supplementary file 1*). These included the nine *Gr* genes previously identified as sugar receptor genes (*Gr5a, Gr43a, Gr61a, and Gr64a-f*) (*Supplementary file 1*), most of which have previously been found to be expressed in legs via *GAL4* driver expression (*Ling et al., 2014*; *Thoma et al., 2016*). Also detected in the leg of *D. melanogaster* were 13 *Ionotropic receptors (IRs)*, many of which have been detected in legs via *GAL4* expression (*Koh et al., 2014*; *Sánchez-Alcañiz et al., 2018*), and 30 *Odorant binding proteins (Obps)* including several previously reported in the leg (*Galindo and Smith, 2001*; *Jeong et al., 2013*; *Figure 4—figure supplement 1*, *Supplementary file 1*).

We compared the leg transcriptomes of *D. melanogaster* and *D. suzukii* with labellar transcriptomes prepared earlier by analogous methods (*Dweck et al., 2021*). A principal components analysis (PCA) showed clear clustering of transcriptomes by organ and by species (*Figure 4A*).

We next performed a pairwise comparison between the leg transcriptomes of *D. suzukii* and *D. melanogaster*. The pan-neuronal gene *nSyb* (*neuronal Synaptobrevin*) and the *IR* co-receptor genes *Ir25a* and *Ir76b* were expressed at similar levels between the two species. Among the nine sugar receptor genes, expression of three (*Gr64a, Gr64d, and Gr64e*) was reduced in *D. suzukii* with an adjusted p-value<0.05 (*Figure 4B and C*, *Supplementary file 3*). None of the sugar *Grs* showed a higher level of expression in *D. suzukii* than *D. melanogaster*. Gr64d was not detected at all in *D. suzukii* (TPM (transcripts per million) = 0, *Supplementary file 2*). The level of *Gr64a* was reduced to 41% of that in *D. melanogaster* (adjusted p-value<0.0001).

To verify the differential expression of the three sugar *Grs* that were found by RNAseq to be expressed at lower levels in the legs of *D. suzukii,* we performed RT-quantitative PCR (qPCR). The reduced expression level in *D. suzukii* was confirmed in all cases (*Figure 4D*).

In the labellum transcriptome of each species, expression of eight sugar *Grs* was detected (*Gr5a, Gr61a*, and *Gr64a-f*), consistent with several previous studies (*Dahanukar et al., 2001*; *Dahanukar et al., 2007*; *Jiao et al., 2007*). Among these *Grs*, expression of seven (*Gr5a, Gr61a, Gr64a, Gr64b, Gr64d, Gr64e,* and *Gr64f*) was reduced in *D. suzukii* with an adjusted p-value<0.05; levels of the other two *Grs* did not differ significantly (*Figure 4E and F*, *Supplementary file 4*). The expression of *Gr64d* was reduced in *D. suzukii* to only 15% of its level in *D. melanogaster* (adjusted p-value<0.001). The reduced expression in *D. suzukii* was confirmed by RT-qPCR for *Gr5a, Gr61a, Gr64a, Gr64d*, and *Gr64e* (*Figure 4G*).

One of the *Gr* genes expressed at lower levels in the *D. suzukii* labellum than in the *D. melanogaster* labellum, according to both RNAseq and RT-qPCR results, was *Gr5a*, which has been identified as a receptor for trehalose (*Dahanukar et al., 2001*). When presented with a choice between 100 mM trehalose and plain medium, *D. suzukii* showed a weaker egg-laying preference for trehalose than *D. melanogaster* (*Figure 5A*). These results suggest that lower levels of *Gr* expression may contribute to the weaker egg-laying preference of *D. suzukii* for sweeter substrates.

We noted that in this experiment with trehalose, *D. suzukii* laid fewer eggs than *D. melanogaster* (*Figure 5B*). This finding suggests that trehalose is a less potent egg-laying stimulus for *D. suzukii* than *D. melanogaster*, which could also result at least in part from lower expression of *Gr5a*. Interestingly, trehalose is a sugar present in yeast (*Jules et al., 2008*; *Jules et al., 2004*), which populate overripe fruits that are oviposition sites for *D. melanogaster* but not *D. suzukii*.

## Certain mechanosensory genes are expressed at higher levels in *D. suzukii*, which prefers harder substrates

In addition to changes in sugar content, fruits undergo changes in stiffness as they ripen. Previous studies have found a difference between *D. suzukii* and *D. melanogaster* in their egg-laying preference for stiff substrates (*Durkin et al., 2021*; *Guo et al., 2020*; *Karageorgi et al., 2017*). We first confirmed and extended the results of these studies and then investigated the possibility that differences in expression levels of mechanosensory channels in the two species could contribute to them.

We performed a no-choice egg-laying assay using agarose plates of differing stiffness, prepared using agarose concentrations ranging from 0.1 to 2%. Measurements with a penetrometer have indicated that ripe strawberries have a stiffness corresponding to agarose concentrations of ~0.6–1.3%, that overripe strawberries have a stiffness corresponding to ~0.7–0.25% or even less, and that early blushing strawberries correspond to as high as 2% agarose or even higher (*Karageorgi et al., 2017*). All plates contained 100 mM sucrose as an egg-laying stimulus. We found that *D. suzukii* laid the fewest eggs on the softest substrate, whereas *D. melanogaster* laid the fewest eggs on the hardest substrate (*Figure 6A*). We then directly compared the preferences of the two species in a two-choice assay. *D. melanogaster* preferred the softer substrate, while *D. suzukii* preferred the harder substrate (*Figure 6B*).

Are there molecular differences in the mechanosensory systems of these species? Taking advantage of the transcriptomes of the legs and the labellum—organs that make direct physical contact with egg-laying sites—we found expression of nine mechanosensory channels genes (*iav, nompC,*

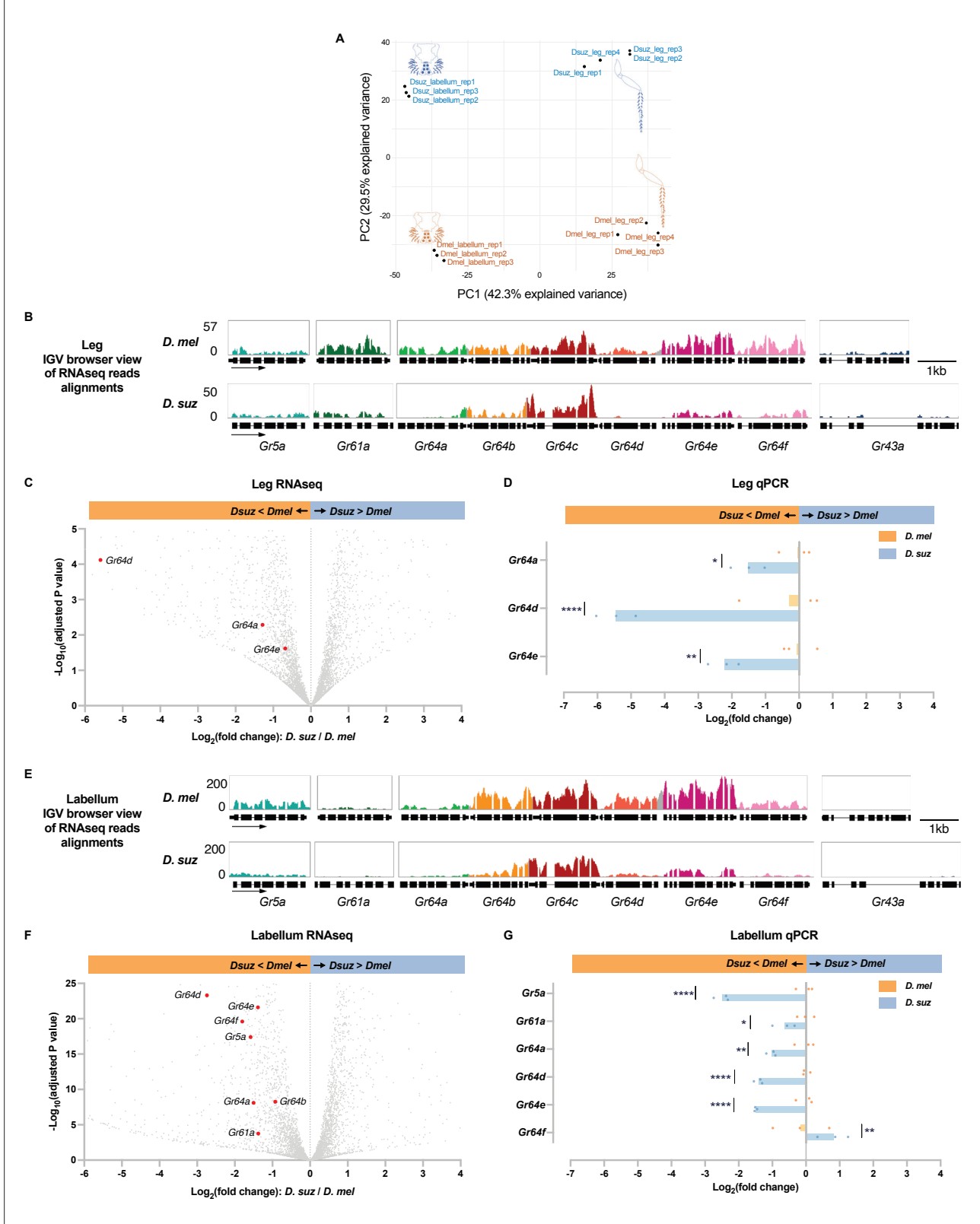

**Figure 4.** Reduced expression of taste receptor genes in the *D. suzukii* labellum and leg. (**A**) Principal component analysis of the labellar and leg transcriptomes of *D. melanogaster* and *D. suzukii*. (**B**) Integrated genomics viewer (IGV) browser view of the aligned reads of the nine sugar *Gr* genes from RNAseq of the legs in both species. Y-axis is adjusted based on the number of mapped reads for qualitative comparison between species. (**C**) Volcano plot of leg transcriptome highlighting differentially expressed sugar *Gr* genes (|log2FC|≥0.58, adjusted p-value<0.05). All other analyzed

*Figure 4 continued on next page*

*Figure 4 continued*

genes with −log$_{10}$ (adjusted p-value) less than 5 and log$_2$ fold-change between –6 and 4 are shown in gray. (**D**) RT-quantitative PCR (qPCR) analysis of three *Gr* sugar receptor genes that were differentially expressed in the RNAseq analysis. Multiple unpaired t-tests are used to compare the expression level between species. n=3. *p<0.05; **p<0.01; ****p<0.0001. (**E**) IGV browser view of the aligned reads of the nine sugar *Gr* genes from RNAseq of the labellum. Y-axis is adjusted based on the number of mapped reads for qualitative comparison between species. (**F**) Volcano plot of labellar transcriptome highlighting differentially expressed sugar *Gr* genes (|log2FC|≥0.58, adjusted p-value<0.05). All other analyzed genes with −log$_{10}$ (adjusted p-value) less than 25 and log$_2$ fold-change between –6 and 4 are shown in gray. (**G**) RT-qPCR results of five sugar *Gr* genes in the labellum. Multiple unpaired t-tests are used to compare the expression level between species. n=3. *p<0.05; **p<0.01; ****p<0.0001.

The online version of this article includes the following figure supplement(s) for figure 4:

**Figure supplement 1.** Gustatory receptor (*Gr*), ionotropic receptor (*Ir*), and odorant binding protein (*Obp*) expression in tibial and tarsal leg segments of *D. melanogaster*.

---

*pain*, *Piezo*, *ppk*, *ppk26*, *Tmc*, *rpk*, and *tmem63*; ≥1 TPM, ***Supplementary file 2***), all of which were expressed in both legs and labellum.

In the legs, six of these mechanosensory genes (*iav*, *nompC*, *pain*, *ppk26*, *Tmc*, and *tmem63*) were expressed at higher levels in *D. suzukii* (***Figure 6C***, adjusted p-value<0.05). Particularly striking was the ~fourfold higher expression of *nompC* (adjusted p-value<0.0001).

In the labellum, levels of four of these mechanosensory genes (*nompC*, *pain*, *Piezo*, and *tmem63*) were again higher in *D. suzukii* (***Figure 6D***, adjusted p-value<0.05). Remarkably, labellar expression of *nompC* was more than sevenfold higher than that of *D. melanogaster* (adjusted p-value<0.0001). None of the mechanosensory genes were expressed at lower levels in *D. suzukii* than in *D. melanogaster*, in either the legs or labellum (***Supplementary files 3 and 4***).

Consistent with these RNAseq results, RT-qPCR analysis revealed higher levels of *nompC* in *D. suzukii* than *D. melanogaster*, in both the legs and labella (***Figure 6E and F***); leg RNA was again prepared from dissected tibia and tarsi of female forelegs. RT-qPCR analysis also confirmed higher expression levels of *Tmc* in the legs and *Piezo* in the labellum.

In summary, whereas *D. suzukii* has a lower preference for sweet and lower levels of sugar receptors, it has a higher preference for stiff substrates and higher levels of certain mechanosensory channels in its legs and labellum.

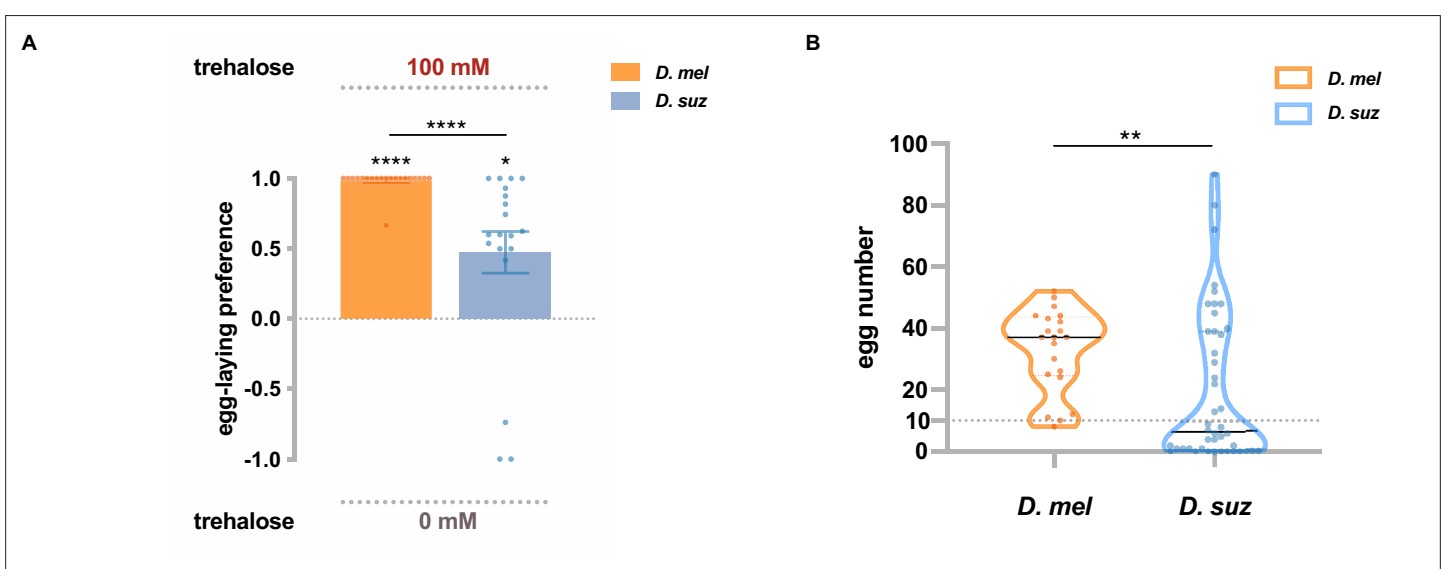

**Figure 5.** Trehalose elicits reduced egg-laying preference and egg-laying in *D. suzukii*. (**A**) Two-choice egg-laying preferences. n=18–20 plates, each with a single fly. Each egg-laying preference index is compared to 0 using the Wilcoxon signed-rank test. The Mann-Whitney test is used to compare the preference indices between species. (**B**) The number of eggs laid on the plates used in (**A**). n=21 plates for *D. melanogaster* and n=44 plates for *D. suzukii*; note that a preference index is calculated only when there are at least 10 eggs on a plate, and thus, the n values in (**B**) exceed those in (**A**). *p<0.05; **p<0.01; ***p<0.001; ****p<0.0001. Error bars are SEM.

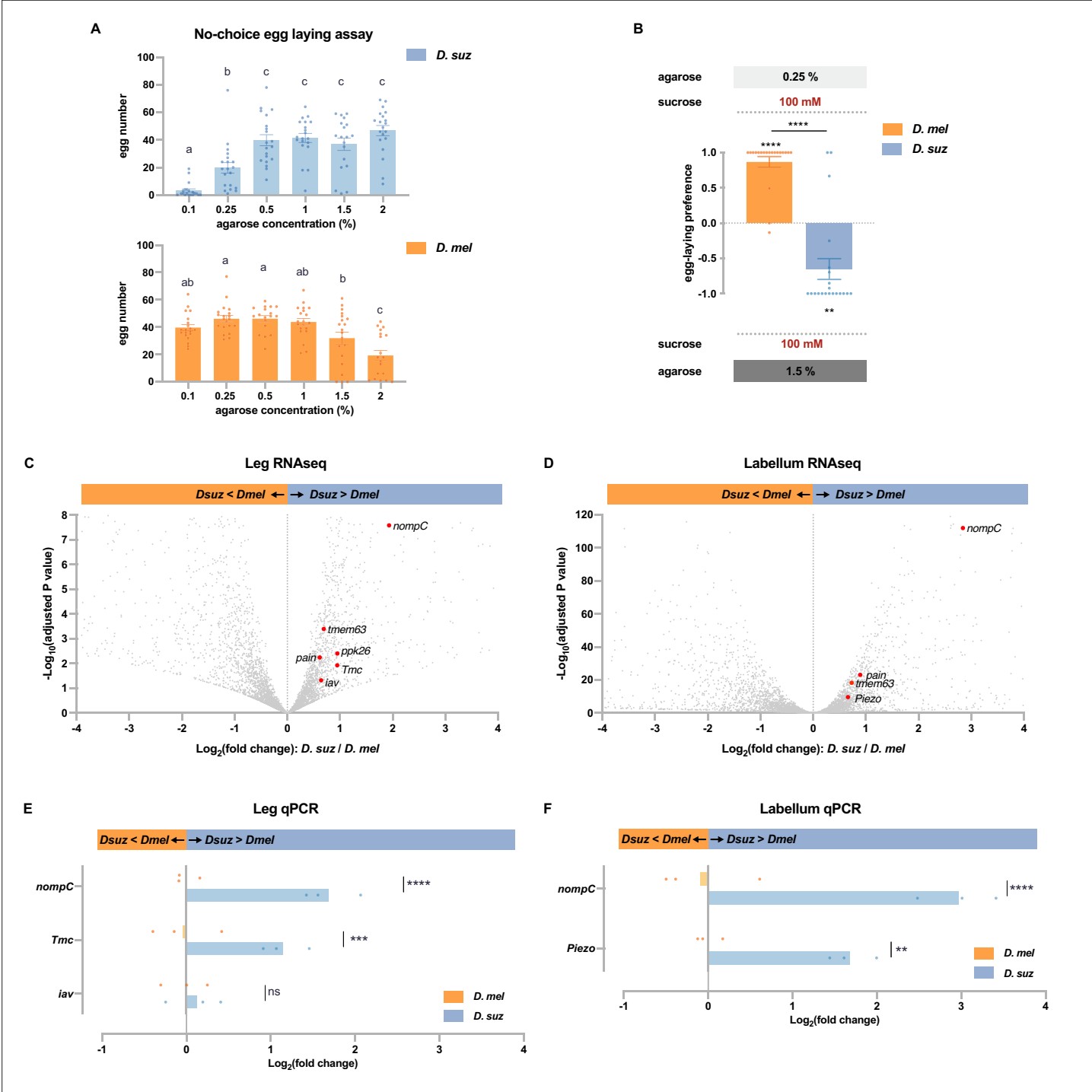

**Figure 6.** *D. suzukii* prefers harder substrates than *D. melanogaster* and expresses higher levels of mechanosensory channels in taste organs. (**A**) Numbers of eggs laid in a no-choice egg-laying paradigm on substrates of the indicated agarose concentrations. n=18–20. One-way ANOVA followed by Tukey's multiple comparison test. Values indicated with different letters are significantly different (p<0.05). Error bars are SEM. (**B**) Two-choice egg-laying preference test for the indicated agarose concentrations. n=18–20. The Mann-Whitney test is used to compare the preference indices between species. n=18–20. **p<0.01; ****p<0.0001. Error bars are SEM. (**C and D**) Volcano plots of leg (**C**) and labellar (**D**) transcriptomes highlighting differentially expressed mechanosensory channel genes. The background gray dots were all other analyzed genes with $-\log_{10}$ (adjusted p-value) less than 8 (**C**) or 120 (**D**) and $\log_2$ fold-change between –4 and 4. (**E and F**) RT-quantitative PCR (qPCR) results for selected mechanosensory channel genes that were found to differ in expression levels between species by RNAseq analysis in leg (**E**) and labellum (**F**). Multiple unpaired t-tests are used to compare the expression level between species. n=3. ns, not significant; **p<0.01; ***p<0.001; ****p<0.0001.

## The two species respond differently to combinations of sweetness and stiffness

The conclusion that *D. melanogaster* and *D. suzukii* have different preferences for sweetness, as well as different preferences for stiffness, raises a question: how do the two species compare in their responses to combinations of sweetness and stiffness? Addressing this question is of interest in part because it may help elucidate principles of sensory integration and in part because it reflects the decisions that flies make in their natural environments. In nature, flies encounter potential egg-laying sites that vary in multiple parameters, and the decisions made by flies of distinct species may be influenced to differing extents by different parameters.

In the previous section (*Figure 6B*), we showed that *D. melanogaster* and *D. suzukii* differed strikingly in their preferences for soft (0.25% agarose) vs. hard (1.5% agarose) substrates in our paradigm, when both substrates contained 100 mM sucrose. We next gave flies a less extreme choice, 0.5 vs. 1% agarose and asked whether their preferences depended on sucrose concentration.

When both substrates contained 100 mM sucrose, *D. melanogaster* showed no preference for the softer substrate (*Figure 7A*). When the sucrose concentration in both substrates was reduced to 30 mM, a preference emerged ($p<0.01$); when the sucrose concentration was further reduced to 10 mM, the preference was again clear ($p<0.001$).

These results support the notion that high sweetness can mask the preference for softness. Our findings are consistent with results found using a different egg-laying paradigm in *D. melanogaster* (*Wu et al., 2019*). The authors of that study speculated that the interaction between taste and mechanosensory input could provide a substrate for evolving different texture selectivity, a notion that can be addressed by testing *D. suzukii*.

We gave *D. suzukii* the same choices of stiffness and again found no preference in the presence of 100 mM sucrose (*Figure 7A*). However, unlike *D. melanogaster*, at lower sugar concentrations, a preference for the softer substrate did not emerge; in fact at the lowest concentration, the flies showed a preference for the harder substrate ($p<0.05$).

These results concern the effect of sugar on the preference for stiffness. We next asked about the effect of stiffness on the preference for sugar. Specifically, we wondered if the dramatic differences between the two species in sugar preferences examined on hard substrates (1% agarose; *Figure 2B*, shown again for convenience as *Figure 7B*) would also be observed on softer substrates (0.5% agarose). They were not, in that the responses of the two species were indistinguishable in all but one case (*Figure 7C*).

We then extended these results by choosing the most dramatic difference between the two species, the preference for 100 mM sucrose vs. 30 mM sucrose at 1% agarose, and asking how this preference changed when the sweeter substrate was also harder. In the case of *D. melanogaster*, the preference for the sweeter substrate vanished (*Figure 7D*). In the case of *D. suzukii*, the opposite result occurred: a preference for the sweeter substrate emerged. In other words, *D. melanogaster* preferred the sweeter substrate unless it was harder. *D. suzukii* preferred the sweeter substrate only when it was harder.

Finally, we aimed to provide a choice between combinations of sweetness and stiffness that may more closely resemble the choice these species make in nature: between a soft overripe fruit that offers access to sugars and a hard ripe fruit whose surface limits access to sugars. Flies may have exposure to higher sugar concentrations in the exposed pulp of certain overripe fruits than on the exterior surface of certain intact, ripe fruits, where they are separated from the interior by a skin. *D. melanogaster* showed a dramatically stronger preference for the softer, sweeter substrate than *D. suzukii* (*Figure 7E*). These results support the conclusion that response to the combination of sweetness and stiffness, along with responses to other sensory cues, contributes to the egg-laying preference shift of *D. suzukii*.

## Discussion

### Differences in sugar sensation between *D. melanogaster* and *D. suzukii*

We have found a constellation of behavioral, physiological, and molecular differences between sugar sensation in *D. suzukii*, which lays eggs on ripe fruit, and *D. melanogaster*, which lays eggs on overripe fruit. These results complement our earlier analysis of bitter sensation in *D. suzukii* (*Dweck et al.,*

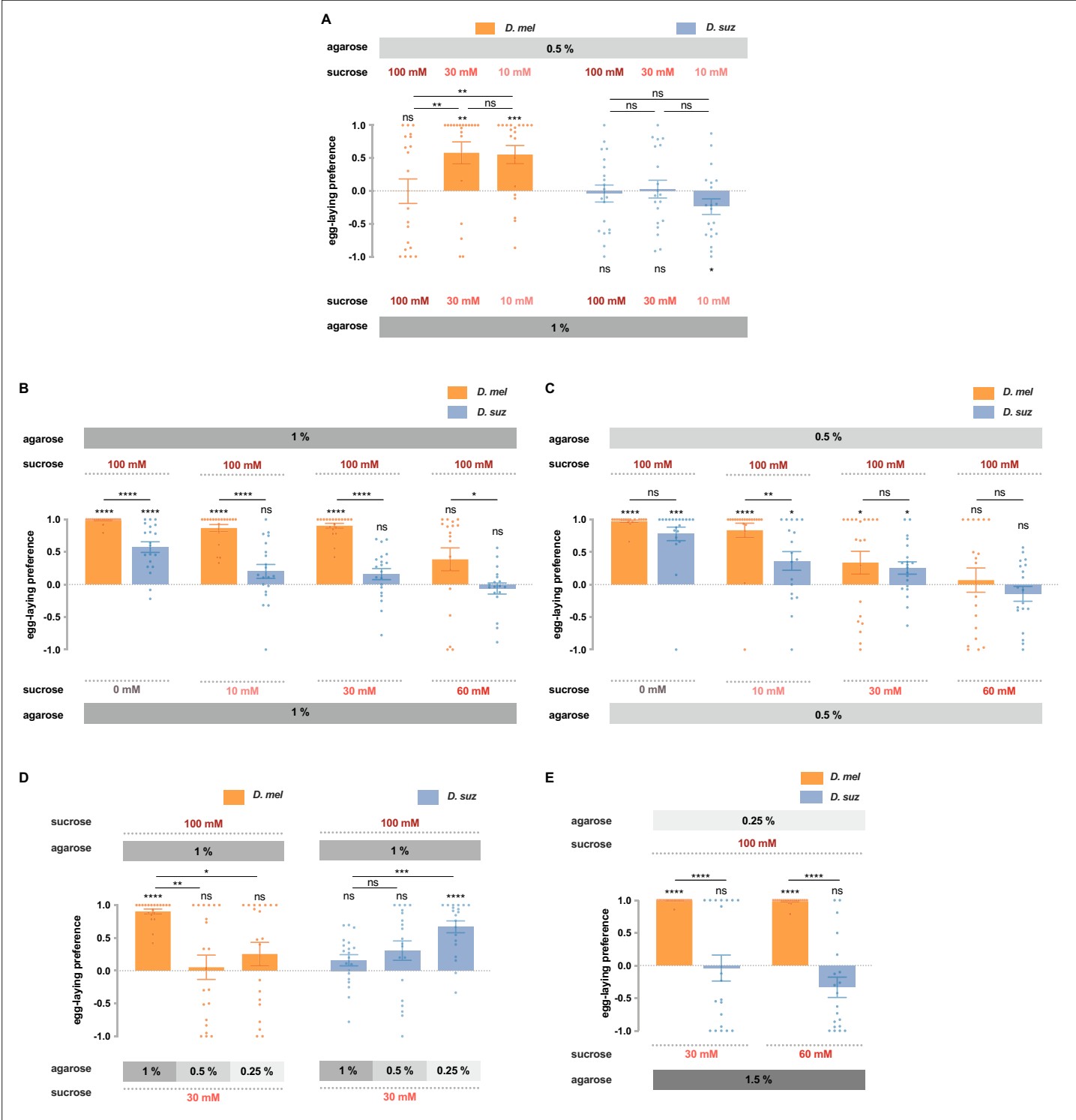

**Figure 7.** *D. suzukii* and *D. melanogaster* respond differently to combinations of sweetness and stiffness. (**A**) The egg-laying preferences of *D. suzukii* and *D. melanogaster* for substrates with the same sweetness but different stiffness. (**B**) The egg-laying preferences for substrates with the same stiffness (1% agarose) but different sweetness (taken from *Figure 2B*). (**C**) The egg-laying preference for substrates with the same stiffness (0.5% agarose) but different sweetness. (**D and E**) *D. melanogaster* and *D. suzukii* respond differently to different combinations of sweetness and stiffness. In D, the first and fourth columns (*D. melanogaster* and *D. suzukii*'s preference for 100 mM sucrose in 1% agarose vs. 30 mM sucrose in 1% agarose) are taken from *Figure 2B* and *Figure 7B*. In all panels, n=18–20, error bars indicate SEM, the Mann-Whitney test is used to compare between two conditions, and the egg-laying preference index is compared to 0 by the Wilcoxon signed-rank test. ns, not significant; p>0.05, * p<0.05, ** p<0.01, *** p<0.001, and **** p<0.0001.

*2021*) and support the notion that major changes in gustation have accompanied the evolution to egg laying on ripe fruit.

*D. suzukii* showed a weaker preference than *D. melanogaster* in each of the 12 tests of sugar preference. Differences between species were found for all three of the primary fruit sugars: sucrose, fructose, and glucose. Several concentration differences were assessed for each sugar. The selected concentrations were intended to simulate choices that the flies make in nature.

We acknowledge that it is difficult to determine with confidence the levels of sugars that flies encounter in their natural environment. Estimates vary with the fruit, the cultivar, and the ripening stage. Different studies have arrived at different conclusions about relative levels of sugar in ripe vs. overripe fruit (*Basson et al., 2010*; *Dudley, 2004*; *Hidalgo et al., 2013*; *Littler et al., 2022*). In any case, sugar concentrations are typically measured in entire, homogenized fruits. In the case of ripe strawberry, sucrose concentrations on the order of 20 mM to 60 mM have been estimated, with glucose and fructose likely ranging from 110 mM to 170 mM (*Lee et al., 2018*; *Paparozzi et al., 2018*). These concentrations are within the dynamic range of taste neurons (*Dahanukar et al., 2007*; *Fujii et al., 2015*; *Hiroi et al., 2004*). However, these interior concentrations are likely to exceed those on the exterior surface of ripe strawberries, which is separated from the interior by a skin.

Consistent with the reduced behavioral preferences, *D. suzukii* shows reduced physiological responses. In the case of sucrose, all S sensilla of *D. melanogaster* show robust responses to a 100 mM concentration. By contrast, most S sensilla on the *D. suzukii* labellum show no response. Leg sensilla of *D. suzukii* responded to sucrose, but dose-response analysis of the f5s sensillum of the leg showed that the response was lower than in its *D. melanogaster* counterpart to higher concentrations of sucrose (*Figure 3—figure supplement 1E*).

Fructose elicited no responses from any labellar sensilla in *D. suzukii* at a 100 mM concentration. By contrast, all L sensilla of *D. melanogaster* responded. Responses of leg sensilla to fructose in *D. suzukii* were similar to those of *D. melanogaster*. We note that the contributions of sugar neurons in the leg have recently been shown to differ from those of the labellum in driving oviposition behavior (*Chen et al., 2022*).

Glucose responses were similar between the two species. The weaker behavioral responses to glucose observed in *D. suzukii* could derive from weaker responses of untested taste neurons. Multiple taste organs, including the pharynx as well as the labellum and legs, contribute to oviposition behavior; sensory neurons of the ovipositor appear to play an important role as well (*Chen et al., 2022*; *Joseph and Heberlein, 2012*; *Yang et al., 2008*). The weaker behavioral response to glucose in *D. suzukii* could also arise from differences in central processing of glucose signals. It will be interesting to determine if there are differences in the connectivity of taste circuits in the two species. Alternatively, taste projection neurons in *D. suzukii* could have a reduced dynamic range, saturate at lower levels of receptor neuron firing, and be less able to distinguish among higher sugar concentrations.

Consistent with the reduced physiological responses, the expression levels of sugar receptor genes were reduced in *D. suzukii*. Particularly, striking was *Gr64d*, whose expression was undetectable in the *D. suzukii* leg and severely reduced in the labellum. Expression of two other sugar receptor genes was also reduced in both organs in *D. suzukii*, as determined by RNAseq and confirmed by RT-qPCR. Four additional sugar receptor genes were found reduced in the *D. suzukii* labellum. We had noted the reduced expression of *Gr64d* in the labellum of *D. suzukii* in our earlier study (*Dweck et al., 2021*), in which we reported genes showing large differences in expression levels (≥fourfold) in an RNAseq analysis but had not confirmed its reduced expression by RT-qPCR.

Although *Gr64d* expression was undetectable in the leg, most of the reductions in *Gr* expression are partial rather than total. However, some Grs may be completely missing from some sensilla, such as those sensilla that show a complete lack of sucrose response in the labellum.

## Differences in mechanosensory gene expression between *D. melanogaster* and *D. suzukii*

While some sugar receptors show a decrease in expression, some mechanosensory channels show an increase. Particularly, striking was *nompC*, which was expressed at higher levels in both legs and labellum of *D. suzukii*. *nompC* is required for the detection of food texture in *D. melanogaster* (*Sánchez-Alcañiz et al., 2017*). Perhaps its greater expression in *D. suzukii*, which could produce either an increase in the number of cells expressing *nompC* or the number of channels per cell, either

of which could in turn produce greater activation of a circuit that contributes to the preference of this species for greater stiffness. Sensory evaluation of stiffness, however, is complex: a study of oviposition preferences in *D. melanogaster* showed a role for *Tmc* in the discrimination of subtle stiffness differences and *Piezo* in the discrimination of mild stiffness differences (*Zhang et al., 2016*). Interestingly, both *Tmc* and *Piezo* are also upregulated in *D. suzukii* (*Figure 6*).

In this evaluation of gene expression, we analyzed hand-dissected taste tissue, specifically the labellum, tarsi, and tibia. However, our results are consistent with those of an RNAseq analysis of whole heads, in that *piezo* was identified in both studies as a gene that was upregulated in *D. suzukii* compared to other species (*Durkin et al., 2021*).

It will be interesting to examine the regulatory architecture of taste and mechanosensory genes that are differentially expressed in *D. suzukii* and *D. melanogaster*. A comparative analysis of their regulatory regions in these and other species might, for example, reveal the loss or gain of enhancer activity in *D. suzukii*. The history of the evolutionary changes we have found could be interesting. Of the three S sensilla that have retained sucrose response in *D. suzukii*, two of them, S3 and S7, are distinct from other S sensilla in their bitter responses (*Dweck et al., 2021*); perhaps S3 and S7 develop via a program that is less vulnerable to the change that eliminated sucrose response in other S sensilla.

An important direction for future investigation will be to determine whether the oviposition and mechanosensory preferences of *D. suzukii* can be altered by increasing the expression of sugar receptors, decreasing the expression of mechanosensory receptors, or by manipulating the activity of the neurons in which they are expressed. We note that in addition to changes in levels and patterns of gene expression, sensory function may also evolve by virtue of changes in the primary sequence of receptors and channels, e.g., *Ir75b* in *Drosophila sechellia* (*Prieto-Godino et al., 2017*).

## Integration of sweet taste and mechanosensation in *D. melanogaster* and *D. suzukii*

Having first examined sweet taste and mechanosensation separately, we then studied them together. We found that *D. suzukii* responds differently than *D. melanogaster* to combinations of sweetness and hardness. Among the principal findings were: (i) when sugar concentrations were progressively reduced, a preference for stiffness emerged in *D. suzukii*, while a preference for softness emerged in *D. melanogaster* (*Figure 7A*); (ii) most of the differences in sugar preference that were observed between the two species at high stiffness were lost at lower stiffness (*Figure 7B and C*); (iii) in a test of sweet preference, *D. melanogaster* preferred the sweeter substrate unless it was harder, whereas *D. suzukii* preferred the sweeter substrate only when it was harder (*Figure 7D*); (iv) *D. suzukii* showed a dramatically lower preference than *D. melanogaster* for substrates that are sweeter and softer, a combination chosen to represent the niche in which *D. melanogaster*, but not *D. suzukii*, prefers to lay eggs.

The different responses of *D. suzukii* to combinations of sweetness and hardness could have evolved via a variety of mechanisms. Recent studies in *D. melanogaster* have identified a number of different receptors, neurons, and mechanisms that may have undergone modification to promote evolutionary shifts in the preference of *D. suzukii*.

First, taste sensilla contain several neurons, most of which are gustatory but one of which is mechanosensory and expresses *nompC*, a gene required for texture discrimination (*Sánchez-Alcañiz et al., 2017*). Activation of this mechanosensory neuron suppresses presynaptic calcium responses of sweet-sensing neurons (*Jeong et al., 2016*). This mechanism could help explain our finding that *D. melanogaster* showed no oviposition preference for 100 mM sucrose vs. 30 mM sucrose when the 100 mM sucrose substrate was much harder, that is, 1 vs. 0.25% (*Figure 7D*). It is conceivable that activation of the mechanosensory neuron by the harder substrate suppressed the sugar neuron, effectively reducing the perceived sweetness and thereby the appeal of the harder substrate to *D. melanogaster*. *D. suzukii*, by contrast, preferred the sweeter substrate even when it was much harder.

Second, a study of egg-laying preferences in *D. melanogaster* supported another mechanism of integration, in which activation of sugar neurons enhances the output of mechanosensitive neurons that express the TMC protein, inhibiting the discrimination of hardness (*Wu et al., 2019*). This mechanism depends on TMC.

Third, besides the mechanosensory neurons located in sensilla, the labellum of *D. melanogaster* also contains a pair of intriguing multidendritic neurons that innervate the base of many taste hairs

(*Zhang et al., 2016*). These neurons are activated by force and orchestrate different feeding behaviors according to the intensity of the force. This pair of neurons also depends on the TMC protein.

It is striking that all three of these mechanisms seem likely to rely on the *nompC* or *Tmc* genes, both of which are upregulated in the taste system of *D. suzukii*, compared to *D. melanogaster*. It is conceivable that the upregulation of these genes contributes to the evolutionary plasticity of circuits that control egg-laying decisions in *D. suzukii*.

While all three of these mechanisms are based on peripheral neurons, there may also be modification of sensory integration in the CNS of *D. suzukii*. We note with interest the identification in *D. melanogaster* of second-order sweet gustatory projection neurons whose presynaptic terminals map to the antennal mechanosensory and motor center in the brain, suggesting the integration of taste and mechanosensory signals at this level as well (*Kain and Dahanukar, 2015*).

In summary, combinations of sweetness and hardness are evaluated differently by the two species. There are a variety of mechanisms that could contribute to this difference, and further studies will be required to delineate whether particular mechanisms have been modified to promote the exploitation of a new niche by *D. suzukii*.

## Evolution of the taste system in the oviposition shift of *D. suzukii*

In a recent study, we found that *D. suzukii* has lost behavioral response to bitter compounds, has lost 20% of the bitter-responding sensilla from the labellum, and has reduced expression of certain bitter-sensitive Gr receptors (*Dweck et al., 2021*). A simple interpretation of the loss of bitter response in *D. suzukii* was that it reduced detection of deterrent bitter compounds in ripe fruit, contributing to a shift toward oviposition on them. In the present study, we have shown that *D. suzukii* also has a reduced behavioral response to sugars, a loss of physiological responses to sugars, and reduced expression of receptors for sugars, relative to *D. melanogaster*.

The reduction in both bitter and sugar responses is consistent with an even simpler interpretation that many of the taste cues that guide the egg-laying decisions of *D. melanogaster* are less salient to *D. suzukii*, as if *D. melanogaster* is more reliant on gustatory information in selecting egg-laying sites. Whereas *D. suzukii* lays eggs in ripe, intact fruits, *D. melanogaster* lays eggs on fruits that vary widely in their degree of decomposition and microbial growth. *D. melanogaster* thus encounters an immense variety of nutrients and toxins while searching for egg-laying sites, and gustation may be critical in evaluating their enormous chemical complexity. There may be great selective pressure on the taste system of *D. melanogaster* to interpret their chemical composition and help distinguish those sites that are most conducive to the survival of offspring.

Consistent with this interpretation, when olfactory, mechanosensory, and visual input were eliminated, *D. melanogaster* showed a stronger egg-laying preference for overripe vs. ripe strawberry purée than *D. suzukii* (*Figure 1B*; note that we expect a ripe purée to contain more sugar than the skin of a ripe fruit, and thus, the preference for overripe fruit may be greater in the field than in this experiment for both fly species). These results support the interpretation that taste cues drive circuits that play a major role in activating egg-laying behavior in *D. melanogaster* but that this role has been diminished in the evolution of *D. suzukii*.

In animal evolution, there are interesting examples of the gain of sweet taste, as in hummingbirds (*Baldwin et al., 2014*), and of the loss of sweet taste, as in cats (*Li et al., 2006*). Sweet taste has been diminished in *D. suzukii* compared to *D. melanogaster* with respect to the parameters considered in this study, but it has certainly not been eliminated: a number of its sensilla show sugar responses (*Figure 3*), and *D. suzukii* prefers to lay eggs on 100 mM concentrations vs. 0 mM concentrations of all three sugars (*Figure 2B–D*). However, when the choices were less extreme, *D. suzukii* did not show a preference, as if it were satisfied with a low concentration and did not distinguish between concentrations above a certain threshold.

There may be selective pressure to retain some degree of sweet taste in *D. suzukii* for several purposes. First, sweet taste may help flies distinguish between ripe fruits and underripe fruits, which may have even less sugar on their skins than ripe fruits. Second, sweet taste may inform other kinds of decisions, including feeding decisions. Sugars are nutritious, and *D. suzukii*, like other flies, needs energy sources.

Our data are also consistent with more complex models for the role of sugar sensation in the shift of oviposition preference in *D. suzukii*. We have directly examined the electrophysiology and receptor

gene expression of peripheral taste organs but not of taste projection neurons or any other neurons in the taste circuit. It is entirely plausible that the primary sensory representation of sugars is transformed in different ways at higher levels in the circuitry of the two species. Our study lays a foundation for further research into the role of sugar sensation in the adaptation of *D. suzukii* to its niche.

In a larger sense, the oviposition decisions of *D. suzukii* are likely driven by a wide variety of cues detected by multiple sensory modalities. Much remains to be learned about the identity and concentration of these cues, as well as about the receptors, neurons, and circuits by which they drive egg laying. Further research into the egg-laying shift of *D. suzukii* may provide insight into mechanisms of sensory system evolution and at the same time have translational implications. Cues that attract *D. suzukii* and promote egg laying could be incorporated into decoys that might contribute to efforts to relieve the burden of this invasive pest on global fruit production.

# Materials and methods

## *Drosophila* stocks

Flies were reared on standard cornmeal-agar medium (Archon Scientific) at 25°C and 60% relative humidity in a 12:12 hr light-dark cycle. *D. melanogaster* Canton-S (CS) flies were used for electrophysiological recordings and behavioral assays. The *D. suzukii* stock was collected in Connecticut (*Dweck et al., 2021*). The *D. melanogaster* sugar *Gr* octuple mutant was from H. Amrein (*Yavuz et al., 2014*).

## Strawberries

Intact fresh strawberries, used in *Figure 1—figure supplement 1A*, were commercially available organic strawberries (Driscoll). Red strawberries with regular shape and uniform color were picked as ripe strawberries. To obtain overripe strawberries, ripe strawberries were kept in a closed plastic bag at 25°C and 60% relative humidity for 48 hr. The overripe strawberries no longer had intact fruit skin, and their pulp was exposed. Whole strawberries were used in this experiment.

Strawberries used to make purées (*Figure 1—figure supplement 1B* and *Figure 1*) were harvested from the Yale Science Building greenhouse. Strawberry plants (*Fragaria ananassa* Duch. cv. Ozark Beauty) were grown in a greenhouse in 6-inch round pots containing ProMix BX with Mycorrhizae. Photoperiod was maintained at 16 hr light/8 hr dark resulting in daily light integrals ranging from 15 to 20 mol/m$^2$/day. Day/night temperatures were 25/20°C, and median humidity was maintained in the range of 30–80% with a median of approximately 50%. Constant fertilization with Jack's 20-10-20 was used to achieve nitrogen levels of 200 ppm.

Full-sized strawberries were harvested. The developmental stages were classified based on color: bright red for ripe strawberries and dark red for overripe strawberries. Strawberries with regular shapes and uniform colors that could be unambiguously assigned to ripe or overripe stages were collected and stored at –20°C without leaves. Strawberry purées were made from these frozen strawberries and stored as 50% w/v purées at –20°C. When making oviposition plates, 1% agarose substrate containing 10% w/v purée of the desired ripening stage was made from the 50% w/v purée.

## Tastants

Tastants were obtained at the highest available purity from Sigma-Aldrich. All tastants were dissolved in 30 mM tricholine citrate, an electrolyte that inhibits the water neuron. All tastants were prepared fresh and used for no more than 1 day.

## Egg-laying assays

The egg-laying assays shown in *Figure 1—figure supplement 1* were performed in cages (24.5 cm × 24.5 cm × 24.5 cm, BugDorm-4E2222, Insect Rearing Cage) that were equipped with two Petri dishes (60 mm × 15 mm, Falcon) containing either a whole ripe strawberry or an overripe strawberry (*Figure 1—figure supplement 1A*) or 1% w/v agarose with 10% w/v purée of ripe or overripe strawberry (*Figure 1—figure supplement 1B*). Newly eclosed flies were maintained in a culture vial supplemented with yeast paste for 5–6 days. About 25 flies (15 females + 10 males; *Figure 1—figure supplement 1A*) or 35 flies (25 females + 10 males; *Figure 1—figure supplement 1B*) were placed in each cage for 24 hr in the dark. The egg-laying preference index was calculated as (egg # on one Petri dish – egg # on the other Petri dish)/(total egg #).

All the other egg-laying assays were carried out in four-quadrant Petri dishes (Dot Scientific, CAT # 557684). Two opposite quadrants contained the concentrations of agarose and concentration of sugars, as indicated in each figure. 10 newly eclosed females were reared with 5 males in a vial for 5 days with yeast paste before the assay. One female fly was placed in each plate except for *Figure 1*. The number of eggs was counted after 48 hr in dark (25°C and 60% humidity). For the preference assays, a very small fraction of dishes contained fewer than 10 eggs in total and were excluded from the results. The egg-laying preference index was calculated as (egg # on one side – egg # on the other side)/(total egg #). We note that very few eggs were laid in the two quadrants lacking agarose or sugars and were not included in the calculation of the preference index.

For preference assays in *Figure 1*, the antennae and maxillary palps of newly eclosed female flies were removed by forceps. In four-quadrant etri dishes, two quadrants contained 1% w/v agarose with 10% w/v purée of ripe and overripe strawberries, respectively. 4–10 females without olfactory organs were placed in each plate. Other details are the same as before.

For no choice assay in *Figure 6A*, two opposite quadrants contained the same substrates, and the total egg number of every plate was counted.

## Electrophysiology

Electrophysiological recordings were performed with the tip-recording method (*Hodgson et al., 1955*), with some modifications; 5–7-day-old mated female flies were used. Flies were immobilized in pipette tips, and the labellum or the female foreleg was placed in a stable position on a glass coverslip. A reference tungsten electrode was inserted into the eye of the fly. The recording electrode consisted of a fine glass pipette (10–15 µm tip diameter) and connected to an amplifier with a silver wire. This pipette performed the dual function of recording electrode and container for the stimulus. Recording started the moment the glass capillary electrode was brought into contact with the tip of the sensillum. Signals were amplified (10×; Syntech Universal AC/DC Probe; http://www.syntech.nl), sampled (10,667 samples/s), and filtered (100–3000 Hz with 50/60 Hz suppression) via a USB-IDAC connection to a computer (Syntech). Action potentials were extracted using Syntech Auto Spike 32 software. Responses were quantified by counting the number of spikes generated over a 500 ms period after contact. Different spike amplitudes were sorted; we did not convolve all neurons into a single value. However, in nearly all recordings in this study, the great majority of the spikes were of uniform amplitude, and those were the spikes whose frequencies we report.

## RNA purification, library preparation, and sequencing

The tarsus and tibia segments of approximately 600 forelegs were hand-dissected from 5-day-old *D. melanogaster* and *D. suzukii* females. Flash frozen segments were ground under liquid nitrogen and resuspended in RLT plus lysis buffer (Qiagen). RNA was extracted using acid phenol and heating at 65°C for 10 min. Residual phenol was removed with chloroform. RNA was then precipitated with isopropanol. Libraries were prepared using KAPA mRNA HyperPrep Kit (Kapa Biosystems) and sequenced on an Illumina HiSeq 2500 or NovaSeq sequencers by the Yale Center for Genome Analysis. Four biological replicates were produced for each species. 30–60 million 75 bp or 100 bp paired-end reads were obtained per sample. Raw reads are accessible at the Genbank SRA database (BioProject accession number PRJNA856346).

## RNA sequencing analysis

Reads were aligned to the *D. melanogaster* genome (BDGP6) and the *D. suzukii* genome (version 1.0) using TopHat (version 2.1.1). *D. melanogaster* leg transcripts were quantified using Cufflinks (version 2.2.2). IGV, Integrative Genomics Viewer (version 2.5.3), was used to inspect the read coverage of genes of interest.

For differential expression analyses, the first reads of each pair were remapped to curated coding sequences (CDS) transcriptomes described by *Dweck et al., 2021*, which here include additional mechanosensory and pan-neuronal genes (*Supplementary file 2*), and counted using HTseq (version 0.6.1). Fold changes were estimated using DESeq2 (version 1.26.0) using ashr for shrinkage (*Stephens, 2016*). The labellar RNAseq dataset used was previously made accessible at the Genbank SRA database (BioProject accession number PRJNA670502; *Dweck et al., 2021*). We report differentially expressed genes with |Log2FC|≥0.58 and adjusted p-value≤0.05.

The PCA plot was generated using the prcomp and ggbiplot packages in R with DESeq2 log-transformed values normalized with respect to library size.

The *D. suzukii Gr64* locus in the current version of the genome contains three gaps. To further analyze the sugar *Gr* genes at this locus, reads were mapped to an improved annotation of this locus obtained by amplifying and sequencing genomic fragments (*Figure 4B and E*).

## RT-qPCR

cDNA was made from 200 ng of labellar RNA as template from using EpiScript (Lucigen). Two biological replicates were prepared per species. RT-qPCR was carried out with iTaq universal SYBR green Supermix (Bio-Rad) using 10 ng of cDNA. Primers were designed to amplify the corresponding regions of *D. melanogaster* and *D. suzukii* cDNAs. In most cases, the same pair of primers was used in both species with two mismatches at most; in all cases, there were no mismatches in the last five bp at the 3' end. For *Gr64d*, no such primers were available, and two different pairs of primers amplifying the same region of the two orthologs were used. Primer efficiency was tested using genomic DNA to verify that comparable amplification was obtained in the two species. Only primers that have similar efficacy based on gel images were used in RT-qPCR. Primers that had abnormal melting curves in RT-qPCR were discarded. *Ir76b* and *nSyb* were used to normalize the expression level of our genes of interest across samples.

Primers used for the reference genes were the following:

> *Ir76b:*
> AAGCACTTTGTGTCCATGCG
> CATGGCAAACGGACAGTGGAC
> *nSyb:*
> TGTGGGCGGACCACACAATC
> AATCACGCCCATGATGATCATCATC

Primers used in *Figure 4* were the following:

> *Gr5a:*
> GTGTTCCCCTACTCCAACTGGC
> CGTCATCCACCTCCCGTATG
> *Gr61a:*
> TTGGTTTTCCTTATCGTGGGCAT
> ACGTTGACCTTTGACCGAAGG
> *Gr64a*
> GGAGGTTGAGCGCCTGATATT
> CTGAAGTCCTTTGCGTCGATTG
> *Gr64d D. mel:*
> TGGCGTATTCGTCAGAATCTG
> GATCACATAGAGCAAACAAAACCAGAAG
> *Gr64d D. suz:*
> TGGCGCATTCGTCAGAATCTG
> GATCACATAAAGCAAACAGAACCAGAAG
> *Gr64e:*
> GAGGTGGACGATGCCATATCC
> GTCAGAGCCACATTGTCCAT
> *Gr64f:*
> GTGTGCCCAAGGAGTCCTGGTG
> GCAGTCCCACAGGTCGTTGTCC
> Primers used in *Figure 6* were the following:
> *iav:*
> ACTTCACCAACGCCATGGAC
> GTCTTCATCGTTTGCTCCACC
> *Tmc:*
> AAGAGCAAATCTTTGAGAACATCCG
> GTGCCGCCATTAAAATTTTAAACCTCG

*nompC:*
AGTGGATGTCTTCGATACGGAA
ATCAGGAATTTCACCAGATGCG
*Piezo:*
ATCAAAATGCATCGGGACAACG
GCGAGGCCAATAACACAAAGG

## Statistical analyses

Statistical tests were performed in GraphPad Prism (version 6.01). All error bars are SEM. $*p<0.05$, $**p<0.01$, $***p<0.001$, and $****p<0.0001$.

## Acknowledgements

We would like to thank Chris Bolick, Nathan Guzzo, and the staff at Marsh Botanical Gardens for their support in maintaining strawberry cultivars and plant growth spaces. This work was supported by NIH grant K01 DC020145 to HKMD; NIH 1F32DC019302 to GJST; NIH grant R35 GM128670 to JMG, and NIH grants R01 DC02174, R01 DC04729, and R01 DC011697 to JRC.

## Additional information

### Funding

| Funder | Grant reference number | Author |
| --- | --- | --- |
| National Institutes of Health | K01 DC020145 | Hany KM Dweck |
| National Institutes of Health | 1F32DC019302 | Gaëlle JS Talross |
| National Institutes of Health | R35 GM128670 | Joshua M Gendron |
| National Institutes of Health | R01 DC02174 | John R Carlson |
| National Institutes of Health | R01 DC04729 | John R Carlson |
| National Institutes of Health | R01 DC011697 | John R Carlson |

The funders had no role in study design, data collection and interpretation, or the decision to submit the work for publication.

### Author contributions

Wanyue Wang, Conceptualization, Resources, Data curation, Formal analysis, Validation, Investigation, Visualization, Methodology, Writing – original draft, Project administration, Writing – review and editing; Hany KM Dweck, Gaëlle JS Talross, Conceptualization, Resources, Data curation, Formal analysis, Funding acquisition, Validation, Investigation, Visualization, Methodology, Project administration, Writing – review and editing; Ali Zaidi, Investigation, Methodology; Joshua M Gendron, Investigation, Methodology, Writing – review and editing; John R Carlson, Conceptualization, Resources, Data curation, Formal analysis, Supervision, Funding acquisition, Validation, Investigation, Visualization, Methodology, Writing – original draft, Project administration, Writing – review and editing

### Author ORCIDs

Wanyue Wang ⓘ http://orcid.org/0000-0002-5378-2873
Hany KM Dweck ⓘ http://orcid.org/0000-0002-7017-5020
Gaëlle JS Talross ⓘ http://orcid.org/0000-0002-4785-0606
Joshua M Gendron ⓘ http://orcid.org/0000-0001-8605-3047
John R Carlson ⓘ http://orcid.org/0000-0002-0244-5180

Decision letter and Author response
Decision letter https://doi.org/10.7554/eLife.81703.sa1
Author response https://doi.org/10.7554/eLife.81703.sa2

## Additional files

### Supplementary files
• Supplementary file 1. FPKM values for *D. melanogaster* leg RNAseq.

• Supplementary file 2. TPM values for all RNAseq samples of *D. melanogaster* and *D. suzukii*.

• Supplementary file 3. DESeq2 differential gene expression analysis of leg RNAseq between *D. suzukii* and *D. melanogaster*.

• Supplementary file 4. DESeq2 differential gene expression analysis of labellum RNAseq between *D. suzukii* and *D. melanogaster*.

• MDAR checklist

### Data availability
Raw reads of newly generated leg RNAseq data are accessible at the Genbank SRA database (BioProject accession number PRJNA856346). The labellar RNAseq dataset used was previously made accessible at the Genbank SRA database (BioProject accession number PRJNA670502). RNAseq data analysis results are included in Supplementary files 1-4.

The following dataset was generated:

| Author(s) | Year | Dataset title | Dataset URL | Database and Identifier |
|---|---|---|---|---|
| Wang W, Dweck HKM, Talross GJS, Zaidi Ali, Gendron JM, Carlson JR | 2022 | Sugar sensation and mechanosensation in the egg-laying preference shift of *Drosophila suzukii* | https://www.ncbi.nlm.nih.gov/bioproject/PRJNA856346 | NCBI BioProject, PRJNA856346 |

The following previously published dataset was used:

| Author(s) | Year | Dataset title | Dataset URL | Database and Identifier |
|---|---|---|---|---|
| Dweck HKM, Talross GJS, Wang W, Carlson JR | 2021 | Evolutionary shifts in taste coding in the fruit pest *Drosophila suzukii* | https://www.ncbi.nlm.nih.gov/bioproject/?term=PRJNA670502 | NCBI BioProject, PRJNA670502 |

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
