## [Editor Report]

The agricultural pest *Drosophila suzukii* displays an interesting and costly evolutionary adaptation of preferring ripe to overripe foods, as compared with the well-studied fruit fly *Drosophila melanogaster*. This study identifies mechanisms that may contribute to this evolutionary shift, including changes in the expression levels of gustatory sweet receptors and mechanoreceptors, and altered electrophysiological responses to sugars. Additional studies involving genetic perturbations in *D. suzukii* are needed in the future to determine the extent to which observed changes contribute to the evolution of substrate preference for egg laying.

---

## [Decision Letter]

**Decision letter after peer review:**

Thank you for submitting your article "Sugar sensation and mechanosensation in the egg-laying preference shift of *Drosophila* suzukii" for consideration by *eLife*. Your article has been reviewed by 3 peer reviewers, one of whom is a member of our Board of Reviewing Editors, and the evaluation has been overseen by K VijayRaghavan as the Senior Editor. The reviewers have opted to remain anonymous.

The reviewers all agreed that the paper was exciting and of potential interest to sensory neuroscientists and evolutionary biologists. Several interesting adaptations were observed in D. Suzuki which could account for behavioral differences in this species. However, it remains unclear whether molecular/physiological changes account for observed behavioral effects. After discussion a couple of additional experiments highlighted below would strengthen links across the paper, and reviewer comments are also pasted in full.

Essential revisions:

1) Please note experiments to explore functional changes in mechanoreceptors as detailed by reviewer #3.

2) See control related to olfactory organ removal of Reviewer #2, comment 1.

*Reviewer #1 (Recommendations for the authors):*

This is a nice, descriptive, and clearly written study which describes behavioral changes in D. Suzuki, and provides possible underlying mechanisms related to changes in receptor expression levels and electrophysiological responses. A major limitation is that causative manipulations are lacking- are the molecular changes observed sufficient to explain the behavioral shift? I realize the challenge of performing manipulation experiments in D. Suzuki, but without such experiments the paper is fairly speculative. There are alternative explanations which are discussed but not investigated- such as changes in sweet-responsive neural circuits that could account for behavioral differences, or receptor mutations that could account for electrophysiological/behavioral differences.

In addition to these salient issues, there are a few smaller improvements that could tighten the paper.

- Do changes in receptor expression account for altered electrophysiological responses? For example, is expression of a fructose receptor lost from L sensilla, or sucrose receptors from particular S sensilla? The persistent glucose responses would seem to indicate that another (perhaps more central as mentioned) adaptation is at play.

- The authors observe identical electrophysiological responses in leg sensilla (Figure 3), despite altered receptor expression (Figure 4). This would seem to call the importance of receptor expression level differences into question, but this should at least be discussed.

*Reviewer #2 (Recommendations for the authors):*

This is an interesting and well-executed study. I only have a few suggestions for improvement.

1) Figure 1B: With their olfactory organs removed, *D. melanogaster* prefer laying eggs on overripe fruit puree, while D. suzukii lay similar numbers of eggs on both the ripe and overripe substrates. Is this indifference by D. suzukii due to the removal of the olfactory organs? Would intact D. suzukii prefer the ripe option? It would be helpful to establish some condition in which D. suzukii prefers the ripe puree over the overripe puree.

2) Another mechanosensory channel expressed in the *D. melanogaster* labellum is tmem63. Was this gene detected in the transcriptome analyses?

3) Supp. Figure 1: Add a legend to the figure for *D. melanogaster* (orange) and D. suzukii (blue).

4) The Discussion wanders a bit. The presentation would be improved by the addition of subheadings.

5) It is noteworthy that the sugar sensitivity of the labellar but not the leg sensilla are reduced in D. suzukii. Some discussion as to why the labella sensilla might be more important for sugar sensation during oviposition would add to the Discussion. Since nompC expression was increased in both labellar and leg sensilla of D. suzukii, this would argue that leg and labella mechanosensory sensilla contribute to the evaluation of hardness during oviposition.

6) The authors mention future directions in the Discussion, which is valuable. In addition, it would be worth pointing out that future experiments aimed at activating and inhibiting sugar and mechanosensory neurons in *D. melanogaster* and D. suzukii, and examining the impact of these manipulations on oviposition might be revealing.

*Reviewer #3 (Recommendations for the authors):*

From the cartoons of the oviposition assays it looks like there are empty quadrants and it's not clear how eggs laid in these empty quadrants are accounted for in the egg-laying preference index. Do females change their participation in the assay by laying fewer eggs?

Figure 1C: The reduction in sugar preference is small and there is still a strong preference overall even in the homozygous mutants. Does this imply that the single remaining GR is sufficient for the preference in *D. melanogaster*? Does this imply that there still may be some leftover olfactory function? It's commonly accepted that removing the olfactory organs ablates olfactory function, but I don't see any controls to confirm that olfactory function is lost here (for example, loss of olfactory preference).

It's likely that changes at multiple levels of sensory processing mediate the altered substrate preferences but it would be useful to directly test the role of expression levels of mechanosensors in mediating stiffness preference. Does genetic up or down-regulation of nompC in *D. melanogaster* alter preference for substrate/tolerance of a higher range of stiffness levels or reduction of levels of mechanosensors in D. suzukii (via RNAi) Although these additional experiments may be outside the scope of this study, the manuscript would benefit from some discussion of how transcript level changes could translate to behavioral preference shift.

The authors show that sugar-sensing Grs show similar response properties in Supplemental Figure 2. It would be informative to compare sequences of *D. melanogaster*/suzukii mechanoreceptors to determine whether there are changes in the coding sequences between these two species would be predicted to affect their function.

Supplemental Figure 1 is missing a legend.

---

## [Author Response]

Essential revisions:1) Please note experiments to explore functional changes in mechanoreceptors as detailed by reviewer #3.2) See control related to olfactory organ removal of Reviewer #2, comment 1.

We greatly appreciate the enthusiasm and helpful comments of the reviewers and Editor. We have addressed the comments as detailed below.

Reviewer #1 (Recommendations for the authors):This is a nice, descriptive, and clearly written study which describes behavioral changes in D. Suzuki, and provides possible underlying mechanisms related to changes in receptor expression levels and electrophysiological responses. A major limitation is that causative manipulations are lacking- are the molecular changes observed sufficient to explain the behavioral shift? I realize the challenge of performing manipulation experiments in D. Suzuki, but without such experiments the paper is fairly speculative. There are alternative explanations which are discussed but not investigated- such as changes in sweet-responsive neural circuits that could account for behavioral differences, or receptor mutations that could account for electrophysiological/behavioral differences.

The reviewer asks an important question: "are the molecular changes observed sufficient to explain the behavioral shift?" Since, as the reviewer notes, it's challenging to address this question in *D. suzukii,* we made it a high priority to address it in *D. melanogaster.* We found that a reduction in the expression of Gr sugar receptors, via genetic deletion, caused a shift in the preference of *D. melanogaster* from overripe toward ripe fruit (Figure 1C). We feel that this result provides strong support for the notion that reduction in the expression of Gr sugar receptors could likewise contribute to a shift in preference toward ripe fruit in *D. suzukii*.

We note that there are a number of other examples in which loss of Gr expression causes a shift in behavioral preferences of *D. melanogaster* (*e.g.* Dahanukar *et al.*, *Nat. Neurosci.* 2001; Dweck *et al.*, *eLife* 2021).

We do not claim that these molecular changes account entirely for the behavioral shift. In fact, we believe that a variety of changes at different levels of organization, ranging from the structure of the receptors to the organizations of neural circuits, are all likely to contribute to the evolutionary shift in preference.

We have added to the Discussion the following statement: "An important direction for future investigation will be to determine whether the oviposition and mechanosensory preferences of *D. suzukii* can be altered by increasing the expression of sugar receptors, decreasing the expression of mechanosensory receptors, or by manipulating the activity of the neurons in which they are expressed."

We hope that when genetic manipulation of *D. suzukii* becomes more convenient that our study will provide a strong foundation for investigation.

In addition to these salient issues, there are a few smaller improvements that could tighten the paper.- Do changes in receptor expression account for altered electrophysiological responses? For example, is expression of a fructose receptor lost from L sensilla, or sucrose receptors from particular S sensilla? The persistent glucose responses would seem to indicate that another (perhaps more central as mentioned) adaptation is at play.

In *D. melanogaster* there is ample precedent for losses of particular receptors that cause losses of electrophysiological response in particular sensilla (*e.g.* Dweck *et al., Current Biology* 2000; Dahanukar *et al., Nat. Neurosci. 2001*).

In *D. suzukii,* it is difficult to attribute losses of electrophysiological responses to reductions in specific receptors. We do not have a receptor-to-sensillum map in this species, so we do not know which receptors are lost from which sensilla.

We hope that our results will contribute to efforts to understand taste coding and its evolution in flies.

- The authors observe identical electrophysiological responses in leg sensilla (Figure 3), despite altered receptor expression (Figure 4). This would seem to call the importance of receptor expression level differences into question, but this should at least be discussed.

The electrophysiological responses to 100 mM concentrations of sucrose, fructose and glucose in the legs of *D. suzukii* look similar to those of *D. melanogaster* in Figure 3M,N,O; however, they are not identical. For example, the dose-response curve in Figure 3—figure supplement 1E shows that the response of *D. suzukii* to sucrose is in fact lower at 60 mM and 100 mM concentrations. We have added a statement to the Discussion clarifying this point: "Leg sensilla of *D. suzukii* responded to sucrose, but dose-response analysis of the f5s sensillum of the leg showed that the response was lower than in its *D. melanogaster* counterpart to higher concentrations of sucrose (Figure 3—figure supplement 1E)."

In a more general sense, some of the spike rates in *D. suzukii* might be higher, and perhaps elicit a different behavioral preference, if *Gr64a, Gr64d,* and *Gr64e* were expressed at the same level as they are in *D. melanogaster*.

Reviewer #2 (Recommendations for the authors):This is an interesting and well-executed study. I only have a few suggestions for improvement.1) Figure 1B: With their olfactory organs removed, *D. melanogaster* prefer laying eggs on overripe fruit puree, while *D. suzukii* lay similar numbers of eggs on both the ripe and overripe substrates. Is this indifference by *D. suzukii* due to the removal of the olfactory organs? Would intact *D. suzukii* prefer the ripe option? It would be helpful to establish some condition in which D. suzukii prefers the ripe puree over the overripe puree.

We have added new results showing that intact *D. suzukii* with olfactory organs prefer the ripe option, in two different assays, with two different sources of strawberry, and with strawberries in two different states of intactness (Figure 1—figure supplement 1).

We note that it is difficult to examine the role of olfaction in a third assay, the single-fly oviposition assay used in Figure 1, because olfactory cues are expected to equilibrate in the small volume of a Petri dish over the long course (48h) of the assay.

2) Another mechanosensory channel expressed in the *D. melanogaster* labellum is tmem63. Was this gene detected in the transcriptome analyses?

We thank the reviewer for raising this point. Yes, *tmem63* is detected in both leg and labellar transcriptomes, and in fact is expressed at higher levels in *D. suzukii* in both cases. We've now added *tmem63* to Figures 6C and D, and have added mention of it to the text. We've also changed "CG11210" to "tmem63" in the Supplementary Files.

3) Supp. Figure 1: Add a legend to the figure for *D. melanogaster* (orange) and D. suzukii (blue).

Thanks; we've now fixed this.

4) The Discussion wanders a bit. The presentation would be improved by the addition of subheadings.

We've now added four subheadings and agree it lends structure to the Discussion.

5) It is noteworthy that the sugar sensitivity of the labellar but not the leg sensilla are reduced in D. suzukii. Some discussion as to why the labella sensilla might be more important for sugar sensation during oviposition would add to the Discussion. Since nompC expression was increased in both labellar and leg sensilla of D. suzukii, this would argue that leg and labella mechanosensory sensilla contribute to the evaluation of hardness during oviposition.

We have added to the Discussion a statement to clarify that although similar, the sugar responses of leg sensilla are in fact not the same in the two species: "Leg sensilla of *D. suzukii* responded to sucrose, but dose-response analysis of the f5s sensillum of the leg showed that the response was lower than in its *D. melanogaster* counterpart to higher concentrations of sucrose (Figure 3—figure supplement 1E)."

We have also added a reference to a recent publication showing that leg and labellar sensilla make distinct contributions to oviposition behavior: "We note that the contributions of sugar neurons in the leg have recently been shown to differ from those of the labellum in driving oviposition behavior (Chen et al., 2022)."

6) The authors mention future directions in the Discussion, which is valuable. In addition, it would be worth pointing out that future experiments aimed at activating and inhibiting sugar and mechanosensory neurons in *D. melanogaster* and D. suzukii, and examining the impact of these manipulations on oviposition might be revealing.

We agree and have added to the Discussion the following statement: "An important direction for future investigation will be to determine whether the oviposition and mechanosensory preferences of *D. suzukii* can be altered by increasing the expression of sugar receptors, decreasing the expression of mechanosensory receptors, or by manipulating the activity of the neurons in which they are expressed."

Reviewer #3 (Recommendations for the authors):From the cartoons of the oviposition assays it looks like there are empty quadrants and it's not clear how eggs laid in these empty quadrants are accounted for in the egg-laying preference index.

Very few eggs were laid in empty quadrants. We've now added a statement to the Materials and methods: " We note that very few eggs were laid in the two quadrants lacking agarose or sugars, and were not included in the calculation of the preference index."

Do females change their participation in the assay by laying fewer eggs?

Yes, in some conditions females laid fewer eggs. We think this phenotype is informative and have included the egg numbers in such cases:

i) In Figure 5B, *D. suzukii* laid fewer eggs than *D. melanogaster* in the experiment with trehalose. This result suggests that trehalose is a less potent egg-laying stimulus for *D. suzukii* than *D. melanogaster*. Both the difference in oviposition preference and the difference in number of eggs laid between the two species could potentially result from the lower expression of *Gr5a* in *D. suzukii*.

ii) In Figure 6A, *D. suzukii* laid the fewest eggs on the softest substrate, whereas *D. melanogaster* laid the fewest eggs on the hardest substrate. These results as well as the results of the two-choice preference assay (Figure 6B) likely reflect the opposing preferences of these two species for substrate hardness. In all other cases, we did not observe differences between *D. suzukii* and *D. melanogaster* in numbers of eggs laid.

We note that in Figure 1C, the *D. melanogaster Gr* octuple sugar mutant without olfactory organs laid fewer eggs than the control. Our interpretation of this reduced egg number is that it arises from the severe loss of chemosensory egg-laying stimuli.

Figure 1C: The reduction in sugar preference is small and there is still a strong preference overall even in the homozygous mutants. Does this imply that the single remaining GR is sufficient for the preference in *D. melanogaster*? Does this imply that there still may be some leftover olfactory function? It's commonly accepted that removing the olfactory organs ablates olfactory function, but I don't see any controls to confirm that olfactory function is lost here (for example, loss of olfactory preference).

We believe that the reason there is still a strong preference overall even in the homozygous octuple sugar receptor mutant is that strawberry purées contain other taste cues besides sugars. We have previously shown (Dweck *et al., eLife* 2021, Figure 2) that the oviposition preference for overripe strawberry purée depends on bitter reception, and we imagine that other taste cues play an important role as well.

The single remaining *Gr* is *Gr43a,* which detects fructose. We tested the octuple sugar mutant in the single-fly egg-laying preference and found that it laid very few eggs (<3 eggs/plate) when given a choice of either 100 mM fructose vs plain medium or 100 mM sucrose vs plain medium. These results suggest that this single remaining sugar *Gr* is not sufficient to drive egg laying on sugar-containing substrates.

We carefully removed both antennae and maxillary palps of flies, discarding those in which even a small part of an antenna or maxillary palp remained; thus we think olfactory function was completely ablated.

It's likely that changes at multiple levels of sensory processing mediate the altered substrate preferences but it would be useful to directly test the role of expression levels of mechanosensors in mediating stiffness preference. Does genetic up or down-regulation of nompC in *D. melanogaster* alter preference for substrate/tolerance of a higher range of stiffness levels or reduction of levels of mechanosensors in D. suzukii (via RNAi) Although these additional experiments may be outside the scope of this study, the manuscript would benefit from some discussion of how transcript level changes could translate to behavioral preference shift.

We agree that it would be useful to test the role of nompC expression levels in mediating stiffness preference, and in fact have initiated a study of this problem. The key experiment is whether reduction in the level of nompC in *D. suzukii* reduces its preference for stiff substrates. Constructing the strains to address this problem rigorously, however, is difficult. We agree with the reviewer's suggestion that this project is outside the scope of this study.

We have added to the Discussion a statement that an increase in nompC expression "could produce either an increase in the number of cells expressing *nompC* or the number of channels per cell, either of which could in turn produce greater activation of a circuit that contributes to the preference of this species for hardness".

The authors show that sugar-sensing Grs show similar response properties in Supplemental Figure 2. It would be informative to compare sequences of *D. melanogaster*/suzukii mechanoreceptors to determine whether there are changes in the coding sequences between these two species would be predicted to affect their function.

We agree that changes between species in the coding sequences of mechanoreceptors could contribute to functional differences, and the revised manuscript. contains a statement to this effect: "sensory function may also evolve by virtue of changes in the primary sequence of receptors and channels, *e.g. Ir75b* in *D. sechellia".* We think that a functional analysis of sequence differences would be an excellent direction of future study.

Supplemental Figure 1 is missing a legend.

Thanks; we've now fixed this.